# Factors Influencing Corporate Social Responsibility Disclosure and Its Impact on Financial Performance: The Case of Vietnam

**Thanh Hung Nguyen** [1],*[ID]**, Quang Trong Vu** [1]**, Duc Minh Nguyen** [2] **and Hoang Long Le** [3]

1   Faculty of Accounting and Auditing, Thuongmai University, Ha Noi 100000, Vietnam;
    trongvuquang@tmu.edu.vn
2   Department of Mathematics, Thuongmai University, Ha Noi 100000, Vietnam; ducminhvcu@tmu.edu.vn
3   Institute of Techniques for Special Engineering, Le Quy Don Technical University, Ha Noi 100000, Vietnam;
    hoanglong1291ktqs@gmail.com
*   Correspondence: thanhhungnguyen@tmu.edu.vn; Tel.: +84-912-670-526

**Abstract:** The study examines the impact of company size, industry sensitivity, government ownership, liquidity and company age on Corporate Social Responsibility Disclosure (CSRD) in 2019 annual reports of listed companies on the Vietnam stock market. We also consider the relationship between CSRD and the financial performance measured by return on assets (ROA) and return on equity (ROE). This study uses descriptive statistics and regression methods to test research hypotheses. The empirical findings show that company characteristics, including firm size, liquidity, government ownership and environmental industry sensitivity, are positively associated with firms' CSRD level. Firm age does not influence the CSRD of listed companies. The CSRD significantly affects both ROA and ROE. Our study provides several suggestions to promote the CSR information disclosure of listed companies and enhance their social responsibility for sustainable development.

**Keywords:** corporate social responsibility disclosure; financial performance; listed company; sustainable development

## 1. Introduction

In recent years, businesses' role and social responsibilities are emerging issues recognized by the community. Businesses that contribute to development are often criticized for being the cause of social problems [1]. The requirement for a full report of social problems such as pollution, waste and depletion of natural resources, quality of products and employees' rights and status in large companies has increased dramatically. Consequently, companies have started to participate in corporate social responsibility (CSR) activities and publish information about them. Many large and older companies, in particular, spend a massive amount of money on disclosing information about their CSR activities. Although CSR has received many enterprises' attention, there is still no general concept of CSR [2]. Most definitions describe CSR as a concept whereby companies combine product, labor, social and environmental concerns with their business operations and interact with their stakeholders voluntarily [3]. Disclosure of CSR is an addition to the financial information system to provide information on business strategies, economic performance, environment and society. CSR information responds to the inquiries of various stakeholders, such as workers, shareholders, investors, clients and authorities. This information reflects society's broader predictions regarding the role of business in the community.

To disclose information, companies usually announce their CSR activities in the annual report or separate social reports such as corporate social responsibility report or sustainability report. However, the reporting indicators and reporting methods are not consistent [3]. Most of the previous studies on CSRD have related to the annual report, which seems to be the essential instrument used by companies to connect their stakeholders [4]. Until now, research on CSRD has mostly been conducted in developed countries such as the

US, the United Kingdom, Japan, Australia, Canada and Germany [5–7], which shows that the number of companies implementing CSR activities is increasing [8,9]. Implementing and disclosing corporate social responsibility information contributes significantly to increasing competitiveness and improving financial performance towards sustainable development. Furthermore, some industries tend to publish more CSR information. For example, consumer-oriented enterprises (banking, services companies) often express their attention to social responsibility related to customers and the community to strengthen their image and increase profits and financial performance [10]. Companies need a more comprehensive approach to understanding the impact of enterprises' social responsibility on stakeholders [11]. The enterprises' social responsibility subsequently improves enterprises' financial performance by improving their communication with related parties.

The purpose of this paper is to explore the factors impacting the decision to disclose CSR information in the annual reports of listed Vietnamese companies and the relationship between CSRD and the financial performance of these companies. The disadvantage of the annual report is that it does not promptly provide social responsibility information as the report is only published after the financial year. Meanwhile, social responsibility information can be disclosed immediately on the website, during the Shareholders Meeting or during the Annual General Meeting. However, we use annual reports to guarantee consistency in measuring CSRD of listed companies in Vietnam. By conducting simultaneous research on the impact of factors on CSRD and the influence of CSRD on financial performance in a financial year, our study's findings propose a new research model of CSRD in developing countries, including Vietnam. This paper focuses on Vietnam's listed companies because of two main reasons. First, research on CSRD is mainly conducted in developed countries, while these studies have not been performed much in developing countries, such as Vietnam. Second, it is important to provide more information on the current situation of CSRD in Vietnam because social responsibility reports of Vietnamese listed companies are elementary and not complied with the standards of Global Reporting Initiative (GRI).

The paper is divided into six sections. The next section introduces the theoretical background, followed by the research hypotheses' development. Section 4 describes the data and methodology. Section 5 presents the results and discussions of the findings. Finally, the conclusion, limitations and future research directions are discussed in Section 6.

## 2. Theoretical Background

CSRD is part of enterprises' critical information disclosure content, showing responsibility to stakeholders for sustainable development activities. Typically, the companies will execute this activity, and the information is announced in their annual report (special publications), such as a sustainable development report. Several theories have been used to explain why companies voluntarily release CSR information in their reports. Empirical studies related to CSRD often use stakeholder theory [12–15] and legitimacy theory [16–22].

As pointed out by Deegan [19], the two theories are closely related and used in a supplemental way. The stakeholder theory refers to the interest of many other objects in a firm's CSR besides the traditional users of accounting information such as shareholders and creditors [14]. They have demand for information regarding the effect of a company's activities on the environment and society. When the company acknowledges the legitimate stakeholders' interests, it will voluntarily report more environmental and social information according to their requirements [23]. Disclosure of social responsibility information builds and improves the image of a socially responsible company, which also satisfies their interests with stakeholders. Some companies expect those good stakeholder relationships to increase financial performance by developing valuable intangible assets such as resources and capabilities [4].

Stakeholder theory also shows that company characteristics (size, foreign ownership, age, industry membership) seem to influence CSRD. Based on the stakeholder's pressure, large enterprises are aware of the importance of CSRD more than smaller companies. Con-

sequently, these enterprises tend to disclose more CSR information than smaller firms [4,24]. Stakeholders in older firms have perceived the role of CSR activities in enhancing financial performance. Numerous previous research has revealed that older firms provide more CSR information than younger firms. Delaney and Huselid [25] found that firm age has a significant impact on levels of CSRD.

Moreover, according to stakeholder theory, CSR information is disclosed differently between sectors. More information in terms of environment, safety and health are reported for the manufacturing sector. Hence, it can be seen that company size, age and type of industry affect the level of social responsibility disclosure under pressure from stakeholders.

The legitimacy theory provides a more comprehensive perspective on CSRD as it acknowledges that social contracts bind firms. Companies agree to take the various actions that society desires to achieve their goals, ensuring their continued existence [8,17,19]. Gray [5] indicates that most of the insights into CSRD originate from this theoretical framework. Public disclosure of environmental and social information is a way of legalizing a company's continued existence or activity to society [5]. Companies with higher age often disclose more CSR information than companies with lower age. On the other hand, Cho and Patten [26] argue that in compliance with legitimacy theory, corporate environmental reporting demonstrates the political and social pressure enterprises face concerning their environmental performance. In their opinion, these pressures can arise from business environments such as cultural, legal and political environments.

Therefore, the company will disclose more information on the environment and society to deal with these pressures and maintain their image of a legitimate company and avoid adverse effects caused by the legitimacy crisis [27]. Companies in sensitive industry sectors such as oil and gas, chemicals, mining and metallurgy, which are commonly considered to negatively affect the environment, will disclose more CSR information than other companies.

## 3. Development of Research Hypotheses

Most previous studies have found the relationship between CSRD and factors belonging to company characteristics such as size, age, type of industry and government ownership [1,5]. Moreover, most of these studies have also shown a relationship between CSRD and corporate financial performance [28–31]. The previous studies are based on some theories such as stakeholder theory and legitimacy theory to clarify these relationships.

### 3.1. Company Size

In the context of legitimacy theory, companies with larger sizes face pressure to disclose information about their compliance with state regulations, helping them access resources of society [10,32]. On the contrary, they are also under pressure and supervision from stakeholders related to enterprises' compliance with labor, resources, community and society. Managers in larger firms will have a higher demand for social information. Therefore, these companies will have to collect and provide more social responsibility information [33]. The studies of Adams et al. [34] in 150 companies in six European countries, Kansala et al. [35] in 80 Indian companies, Neu et al. [36] in 33 Canadian public companies, Suwaidan et al. [37] in 65 Jordanian industrial companies and Tagesson et al. [38] in 267 Swedish industries and listed firms also asserted the positive relationship between company size and amount of information published regarding social responsibility practices. Larger firms with a comprehensive operation area have more extensive and more diverse stakeholders [39]. Moreover, larger firms will realize better practice social responsibility, considering that social responsibility and disclosure are a way to enhance the company's reputation and image. Therefore, we can form our first hypothesis:

**Hypothesis 1 (H1).** *There will be a positive relationship between company size and CSRD.*

### 3.2. Industry Sensitivity

Along with size, industry sensitivity is the most common factor impacting CSRD [5,10,34]. These studies show that enterprises operating in the manufacturing sector often have adverse effects on the environment. As a result, they must report more environmental and labor information than enterprises in other industries [3]. The failure or delay in releasing environmental information may signal to the company's stakeholders that it is not following environmental requirements, which can lead to business risks. In general, firms operating in environmentally sensitive industries such as mining, oil, metallurgy and chemicals tend to publish more environmental information than companies in less sensitive industries [1,14,26,40]. Meanwhile, financial and service industries are more likely to report social issues and philanthropy [41,42].

This discrepancy also relates to stakeholder pressure and regulation in the service industry. Firms operating in this sector are subject to considerably less pressure in terms of environmental performance. As a result, they show a less active level of information disclosure [3]. Research of Newson and Deegan [43] in the 149 largest listed commercial and industrial firms by market capitalization were selected from Australia, Singapore and South Korea, providing additional evidence of the relationship between operational industry and CSRD. Wanderley et al. [44] mentioned the relationship between industry membership and CSRD in 127 most prominent companies from emerging countries, such as Brazil, China, India, Indonesia, South Africa and Thailand. The findings showed that CSR information disclosure is related to the industry sector. Enterprises disclose information about their production efficiency and firm performance in environmental and social aspects towards sustainable development goals. The previous results show that companies operating in different sectors have different levels of CSRD. Therefore, we propose the hypothesis that:

**Hypothesis 2 (H2).** *There will be a positive relationship between industry sensitivity and CSRD.*

### 3.3. Government Ownership

Many researchers have pointed out government ownership as another influencing factor in the social and environmental disclosure process [38,45–47]. Cormier and Gordon [46] surveyed three major power companies in Canada to prove that ownership impacts the extent of social and environmental disclosure. The empirical findings showed that two publicly owned companies are more likely to disclose information than one private firm. According to the authors, using legitimacy theory, state-owned and large-sized companies that receive political support must present more information due to accountability and visibility reasons. In contrast, a study by Secchi found that state-owned firms provide less information than private-sector firms [47]. Tagesson et al. [38] investigated firms in Sweden, and found that state firms disclose more social and environmental information. The authors argued that these firms often have more pressure from the government and media about their environmental and social impacts. They also pointed out that Sweden's national culture is the factor that leads to the release of much information regarding social and environmental impacts. This result which can be explained as public sector transparency is a long-standing Swedish tradition. Through transparent and accountable reporting, organizations can strengthen stakeholders' trust in the business activities contributing to sustainable development.

In Vietnam, there is a large number of listed firms with a certain percentage of government ownership. These state-owned enterprises are expected to be typical examples of compliance with disclosure. Thus, state shareholders and their interests are expected to be addressed by releasing social responsibility information in response to government requests as part of a designed strategy. Based on data from 133 enterprises in Malaysia, Amran and Dive [45] suggest that state ownership positively impacts CSRD. Therefore, our expectation of the relationship between state ownership and CSRD is positive, and the following hypothesis is formally stated:

**Hypothesis 3 (H3).** *Government ownership will have a positive relationship with CSRD.*

### 3.4. Liquidity

Abd-El Salam and Weetman [48] show that the higher liquidity leads to a higher level of voluntary CSRD. Ezat and Em-Masry [49] provide evidence of a positive relationship between firm liquidity and online reporting. Similarly, Samaha and Dahawa [50] confirmed a positive relationship between CSR and voluntary social responsibility disclosure. Firms with higher liquidity have excellent financial performance and will release more CSR information. Considering that liquidity may increase access to new business opportunities, that company must disclose more social information [51]. We expect that businesses with high liquidity ratios will voluntarily publish their CSR. Therefore, we formulate our fourth hypothesis as follows:

**Hypothesis 4 (H4).** *There will be a positive association between liquidity and CSRD.*

### 3.5. Company Age

From the perspective of legitimacy theory, the company's renown is built with age. Once established and grown, corporate social responsibility practices tied to their brand may become more critical. Stakeholder expectations regarding firms continue to carry out and disclose corporate social responsibility information because long-term experience helps them use resources effectively and protect the business's reputation with social responsibility actions. [12]. Company age was used in several prior studies as a critical factor influencing social responsibility disclosure [25,52–55]. Research by Delaney and Huselid [25] shows that the older a business is, the higher the level of social responsibility disclosure is. Similar results are found in another study by Kansala et al. [35], which surveyed 80 businesses in India to investigate the extent to which socially responsible information is disclosed. In contrast, some studies find a negative relationship between CSRD and the number of years in business. For instance, Rettab et al. [55] report a negative association between CSR and firm age. Liu and Anbumozhi [54] also report a negative relationship between environmental disclosure and company age. Hence, based on the above points, we set the fifth hypothesis as below:

**Hypothesis 5 (H5).** *There will be a positive relationship between company age and CSRD.*

### 3.6. Financial Performance

Based on stakeholder theory, some prior studies suggest a relationship between social responsibility disclosure and financial performance [29,31,56]. Uwuigbe and Egbide [56] research the relationship between disclosing social responsibility information in Nigerian enterprises with their financial performance measured by return on assets (ROA) and returns on equity (ROE). Their research results show that there is a positive relationship between CSRD and company financial performance in Nigeria. Moore and Robson [29] examined eight supermarket firms to analyze the effect of CSRD on financial performance. These were based on the derivation of a four-measure financial performance index. Their results show that these two variables have positive and significant relationships. Malik and Kanwal [31] examined the relationship between CSRD and financial performance in Pakistan Pharmaceutical firms using ten years of annual report data from 2005 to 2014. The empirical findings demonstrate that CSRD has a significant and positive relationship to ROA and ROE in both directions. In the ASEAN region, Ratmono et al. [57] studied 194 listed companies in Indonesia during the 2015–2017 period and found that CSRD had a significant positive effect on the companies' financial performance. In addition, Saleh et al. [58] examined the 200 highest market capitalization companies on the Kuala Lumpur Stock Exchange (KLSE) from 2000 to 2005 using ROA to measure financial performance. The result indicated a relationship between CSR information and financial

performance, suggesting that these firms can achieve advanced levels of financial performance if they engage in social and environmental activities.

However, according to the legitimacy theory perspective, CSRD can positively or negatively impact financial performance and profitability [36]. Some authors who find a negative relationship [28,30,51] believe that disclosing CSR is stakeholders' disadvantage because a corporation must use its resources only to maximize its profits. Preston and O'Bannon [28] indicate that better corporate social responsibility practices cause lower financial performance. This hypothesis is supported because when firms perform social responsibility, they have to expend many resources. Therefore, the company's profit-maximizing goals are affected. Another study by Andrian [59] on the relationship between CSRD and financial performance was conducted for the consumer goods enterprises listed in the Indonesia Stock Exchange (IDX) between 2015 and 2017. The author analyzed multiple regression on 114 purposive selected observations and reported that CSRD had a significant negative impact on financial performance. Based on the theoretical framework, to analyze the relationship between CSRD and financial performance, we hoped that CSRD would positively impact financial performance. Thus, we hypothesize the following:

**Hypothesis 6 (H6).** *There will be a positive relationship between CSRD and financial performance.*

Given the above research hypotheses, the theoretical framework was developed as shown in Figure 1.

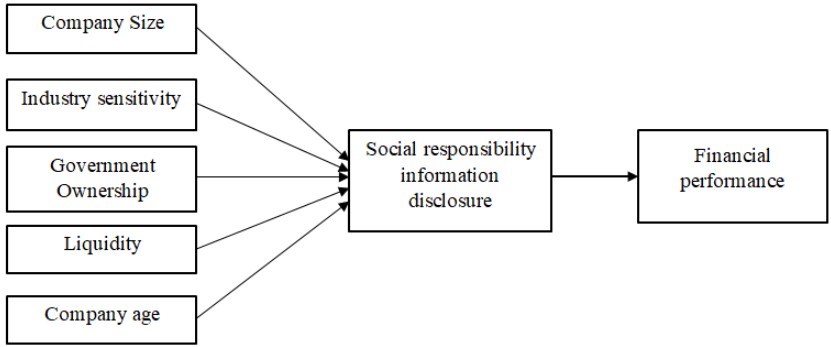

**Figure 1.** Theoretical framework of the study [34,35,38,45,56].

## 4. Data and Methodology

### 4.1. Sample

Our samples were listed companies on the Vietnam Stock Exchange as of 31 December 2019. These enterprises are required to publish information on corporate social responsibility following the regulations of the State Securities Commission of Vietnam. Social responsibility information is disclosed in the content of the annual report. In this study, we collected information from the annual report in 2019, because annual reports in 2020 are published during the period from April to May of 2021. The annual reports were collected by entering the company web pages or by downloading them from the Ho Chi Minh Stock Exchange's website.

As of 31 December 2019, Vietnam had 1575 listed companies. Due to our inability to collect information of all enterprises, we chose Vietnam's top 100 listed companies (called VN100). VN100 is a group of 100 listed companies, including 30 type-one enterprises (called VN30) and 70 type-two enterprises (called VNMidcap) with market capitalization accounting for about 90% and representing more than 80% of the transaction value of the whole market. The sample encompassed 22 industrial companies, 18 real estate companies, 17 banking and insurance companies and 11 food, beverage and tobacco companies and other sectors (Table 1)

**Table 1.** Distribution of the usable sample by industrial classification.

| No | Sector | Number of Companies | Percentage |
|----|--------|---------------------|------------|
| 1 | Information Technology | 2 | 2.0 |
| 2 | Industrials | 22 | 22.0 |
| 3 | Materials | 9 | 9.0 |
| 4 | Financial | 17 | 17.0 |
| 5 | Utilities | 7 | 7.0 |
| 6 | Energy | 2 | 2.0 |
| 7 | Communication Services | 1 | 1.0 |
| 8 | Consumer Discretionary | 8 | 8.0 |
| 9 | Health Care | 3 | 3.0 |
| 10 | Food, Beverage and Tobacco | 11 | 11 |
| 11 | Real Estate | 18 | 18 |
| | Total | 100 | 100 |

*4.2. Measurement*

4.2.1. Company Size

Dimensions of a company can be defined by several measurements, such as turnover, sales, revenues, total assets and number of employees. Following prior research, size is computed as the natural logarithm of total assets, as reported in the Financial Position Statement in 2019 [33,54,60].

4.2.2. Industry Sensitivity

In our research, "more sensitive" sectors are examined to have community concern because their activities involve a higher risk of environmental effect. As presented in the literature, businesses in the industries mining, oil and gas, construction and building materials, chemicals, forestry and paper, steel and other metals, electricity, gas distribution and water are identified as "more sensitive" sectors. The remaining fields are considered as "less sensitive". A one/zero variable is used to assign companies from these industries— one if the company is from a more sensitive industry and zero if it is from a less sensitive industry [3,4].

4.2.3. Government Ownership

An enterprise in which the government holds more than 50% of charter capital is called a government enterprise (Law on Enterprises in Vietnam, 2020) [61]. This variable is measured from data relating to significant shareholders' ratio in annual reports of listed companies. Thus, if the government ownership ratio (measured by the percentage of government ownership in a company) is >50%, we designate it a value of 1, and if not, it is designated with a value of 0 [3,23].

4.2.4. Liquidity

Liquidity is measured by the current ratio, which is counted by current assets/current liabilities in the Financial Position Statement in 2019 [51,60].

4.2.5. Company Age

We measured company age by the number of years since establishment as of the end of 2019 [54,55].

### 4.2.6. Financial Performance

The accounting approach is used to measure financial performance by Return on Assets (ROA) and Return on Equity (ROE). ROA is an indicator of how profitable a company is relative to its total assets, calculated by net profit over average total assets (ROA = net profit/average of total assets). ROE is also another essential financial ratio related to its profitability, which is calculated by dividing a company's net income by its shareholders' equity (ROE = net profit/average of total equity capital).

### 4.2.7. Corporate Social Responsibilities Disclosure

The annual reports of companies in VN100 listed on the Vietnam Stock Exchange, classified in Table 1, were used to calculate CSR disclosure scores and collect the required information to measure CSRD. The annual report of the companies was used because of the following reasons. First, according to the provisions of Circular 155/2015/TT-BTC [62], a legal document of the Vietnamese government, listed companies must prepare and publish their annual reports. Second, the difference in publishing annual reports between enterprises is 4 to 5 months after the fiscal year-end date. Finally, annual reports make it easier to compare companies than other communication channels.

We use the CSR disclosure scores index to measure CSR disclosure level by collecting information from Section 6, "Report the enterprise's impact on the environment and society", of the annual report. A "yes/no" or (1, 0) scoring methodology was employed [4]. Companies that present the information in Section 6 of the report are encrypted as 1. In contrast, companies that do not publish any information are coded as 0. The maximum points are 15, corresponding to 15 categories in the checklist, including 9 environmental items, 5 employee items and 1 society item (Table 2). The level of each company's CSRD is measured as the ratio of its disclosure score to the maximum points possible to achieve. The disclosure index is expressed as a percentage. Consequently, the formula to compute the CSRD index is as below:

$$CSRDI_i = CSR_i/M \tag{1}$$

where

$CSRDI_i$: CSR disclosure index of company i;
$CSR_i$: total disclosure score of company i;
M: maximum score of items (15).

**Table 2.** Items specified in the Vietnamese annual report.

| Aspects | Item Examples |
|---|---|
| Environment: Management of raw materials | EN1: The total amount of materials used in the production and packaging of the company's main products and services during the year |
| | EN2: The percentage of materials recycled to manufacture products and services of the company |
| Environment: Energy consumption | EN3: Energy consumption—directly and indirectly |
| | EN4: Energy saved through innovative energy efficiency |
| | EN5: Energy efficiency Initiative Reports and report on the results of these initiatives |
| Environment: Water consumption | EN6: Water supply and amount of water used |
| | EN7: Percentage and total volume of recycled and reused water |
| Environment: Compliance with the law on environmental protection | EN8: Number of times fined for not complying with laws and regulations on the environment |
| | EN9: The total amount fined for not complying with laws and regulations on the environment |

**Table 2.** *Cont.*

| Aspects | Item Examples |
|---------|---------------|
| Employees: Policies related to employees | EM1: Number of workers |
| | EM2: Average wages of employees |
| | EM3: Labor policies to ensure health, safety and welfare of employees |
| | EM4: The average number of training hours per year |
| | EM5: The skills development program and continuous learning to support workers to ensure employment and career development |
| Society: Responsibility for local community | SO1: Community investment and other community development activities, including financial support to serve the community |

Source: Circular No. 155/2015, Ministry of Finance, Vietnam.

In this study, we did not differentiate the importance between indicators related to the environment, employees and society. Similar to previous studies, this approach emphasizes the number of occurrences of socially responsible disclosures, without regard to the quality, scope, depth and length of these disclosures [23,26,54].

*4.3. Empirical Models*

Statistical analysis performed in this study includes the use of linear regression models to analyze the relationship between CSRD and influencing factors such as company size, industry sensitivity, government ownership, liquidity and company age. At the same time, the study will test the relationship between CSRD and financial performance with ROA and ROE. The following summarizes the three estimated models:

$$CSRD = \beta_0 + \beta_1 \, SIZE + \beta_2 \, INDS + \beta_3 \, GVO + \beta_4 \, LIQ + \beta_5 \, AGE \qquad (2)$$

$$ROA = \beta_0 + \beta_1 \, CSRD \qquad (3)$$

$$ROE = \beta_0 + \beta_1 \, CSRD \qquad (4)$$

where

CSRD: level of corporate social responsibilities disclosure;
SIZE: company size;
INDS: industry sensitivity;
GOV: government ownership;
LIQ: liquidity;
AGE: company age;
ROA: average return on assets;
ROE: average return on equity.

## 5. Results and Discussion

*5.1. Descriptive Analysis*

Based on the results presented in Table 3, social information is presented with the most significant percentage (87%). The employee information accounts for 65% of the information presented by enterprises. Environmental information has the lowest rate of disclosure by enterprises (31.7%). The results also indicate that employee data is important information, with 82% of firms presenting. Most companies have a disclosure for the number of employees. The percentage of labor policies to ensure the health, safety and welfare of workers is 74%.

**Table 3.** Descriptive analysis.

| Item | Percentage (%) | Average Percentage (%) |
|---|---|---|
| EN1 | 28 | |
| EN2 | 22 | |
| EN3 | 36 | |
| EN4 | 32 | |
| EN5 | 24 | Environment: 31.7 |
| EN6 | 35 | |
| EN7 | 21 | |
| EN8 | 45 | |
| EN9 | 42 | |
| EM1 | 82 | |
| EM2 | 55 | |
| EM3 | 74 | Employees: 65 |
| EM4 | 49 | |
| EM5 | 65 | |
| SO1 | 87 | Society: 87 |

Note: Table 3 describes the percentage of the total sample of information disclosure in each item.

In contrast, the environment information only reached 31.7%. Although companies announce nine environmental items, the level of disclosure of these indicators is low (<50%). EN7 is at the lowest level, with disclosure percentages amounting to 21%, followed by EN2 "The percentage of materials recycled to manufacture products and services of the company" and EN5 "Energy efficiency Initiative Reports and report on the results of these initiatives" at 22% and 25%, respectively. Currently, companies seem to have not put weight on preserving and protecting the environment.

In the direction of sustainable development, in 100 annual reports of listed companies, 45 enterprises mentioned sustainable development orientations through promoting and making transparent disclosure of social and environmental information for stakeholders.

*5.2. Analysis of the Main Results*

The research hypotheses are tested by the multiple regression method. The autocorrelation test, heteroscedasticity test and multicollinearity test are conducted before analysis to ensure the reasonableness of conclusions drawn based on multiple regression results. The possible existence of multicollinearity is tested based on the correlation matrix, incorporating all the independent variables and the variance inflation factor (VIF). First, we assess the factors influencing corporate social responsibility disclosure, so the article examines the multicollinearity phenomenon between the first regression equation variables. The multicollinearity test results of the model show that the coefficient VIF < 2. Thus, there is no multicollinearity phenomenon in our study.

Next, Table 4 indicates the correlation coefficients among independent variables. The correlation coefficient matrix results showed five variables, SIZE, INDS, GOV, LIQ and AGE, with correlation coefficients between from 0.0067 to 0.401 (<0.6), statistically significant at a 1% level. It can be seen that when the independent variables have a linear correlation, the multicollinearity phenomenon is less likely to appear.

**Table 4.** Correlation coefficients among independent variables.

| | SIZE | INDS | GOV | LIQ | AGE |
|---|---|---|---|---|---|
| SIZE | 1 | 0.258 ** | 0.401 ** | 0.377 ** | 0.272 ** |
| INDS | | 1 | 0.266 ** | 0.117 ** | 0.067 ** |
| GOV | | | 1 | 0.273 ** | 0.308 ** |
| LIQ | | | | 1 | 0.214 ** |
| AGE | | | | | 1 |

Note: ** Significant at the 0.01 level (two-tailed).

Multiple regression methods are undertaken to test the study's hypotheses. The regression models relating to CSRD showed that size, industry sensitivity, government ownership and liquidity have a relationship with CSRD (Table 5), being statistically significant at the 5% level. The adjusted $R^2$ suggests that approximately 42% of the variation in the CSRD can be explained by the independent variables included in these regression models.

**Table 5.** Results of the regression models for CSRD.

| Variable | Coefficient | Std. Error | t-Statistic | Prob. | VIF |
|---|---|---|---|---|---|
| C | −0.598515 | 0.216236 | −2.767880 | 0.0068 | |
| SIZE | 0.822513 | 0.217217 | 3.786590 | 0.0003 | 1.119416 |
| INDS | 0.549640 | 0.116457 | 4.719666 | 0.0000 | 1.112852 |
| GOV | 0.347067 | 0.117240 | 2.960313 | 0.0039 | 1.053056 |
| LIQ | 0.093069 | 0.041803 | 2.226393 | 0.0284 | 1.010602 |
| AGE | 0.064111 | 0.045107 | 1.421295 | 0.1585 | 1.028219 |
| R-squared | 0.449939 | | | | |
| Adjusted R-squared | 0.420680 | | | | |

Note: Table 5 presents results of the regression models to test hypothesis H1–H5.

In Hypothesis 1, the coefficient for the path from SIZE to CSRD is positive and significant ($\beta = 0.822$, $p < 0.01$). The hypothesis H1 is accepted, which means company size would have a significant effect on CSRD. Table 5 shows that larger companies release more information on their social responsibility practices. This result is also consistent with the research of Brammer and Pavelin [39] for the UK, Branco and Rodrigues [4] for Portuguese listed companies and Reverte [3] for Spanish listed firms. This can be explained by the likelihood that large companies will be more interested in stakeholders and society, meaning CSR information related to the company is highly demanded, thereby creating a certain pressure on CSRD. In addition, large-scale companies in Vietnam (total assets over USD 1 million) will have abundant financial resources and more professionals to employ in information disclosure. They have departments to collect, process and provide social responsibility information. Therefore, their firm's CSRD index will be higher than that of smaller-scale companies. As a result, CSR information in larger firms is more transparent than in smaller firms.

In Hypothesis 2, we expected a positive relationship between industry sensitivity (INDS) and CSRD. With $\beta = 0.549$, $p < 0.01$, hypothesis H2 is supported. The results of this research are also consistent with the study of Hackston and Milne [1] in New Zealand, Brammer and Pavelin [40] in the United Kingdom and Reverte [3] in Spain. Legitimacy theory is considered the most appropriate theory to explain CSRD activities of Vietnamese listed companies. Vietnamese firms have to produce reports on CSR activities in response to community pressure and build or sustain corporate legitimacy. Listed manufacturing companies are environmentally sensitive industries, especially 22 industrials companies, 11 food, beverage and tobacco companies and 2 energy companies, which tend to publish more social information—including environment information, labor information and community information—than firms less sensitive to the environment. In this respect, CSRD can be seen to build and enhance a company's competitiveness.

The coefficients of government ownership and liquidity are also positive. When these variables' value increases, the firm's CSRD index also increases correspondingly. Thus, it supports the research hypothesis about the relationship between CSRD and government ownership (H3). The result of the study is not only consistent with results of research in developed countries such as Canada [46] and Sweden [38] but also complies with previous studies in developing countries such as Malaysia [45] and Tunisia [63]. State-owned enterprises can be seen receiving a lot of social expectations as well as attention from the community. These firms will have stricter scrutiny and will disclose more social responsibility information. In Vietnam, state-owned enterprises are the leading enterprises in the industry, with large-scale and abundant financial resources, involving many localities

and employing many employees [64]. Therefore, state-owned enterprises tend to disclose complete CSR information. This is also consistent with our first hypothesis that company size has a positive effect on CSRD. In addition, hypothesis H4 indicates the weak linkage between liquidity and CSRD with a small effect β = 0.093. With *p*-value < 0.05, H4 is accepted, complying with the studies of Waddock and Graves [51] and Hussainey et al. [60].

Company age is not significant for the effects on CSRD, with level sig. = 0.158 > 0.05, which does not support Hypothesis 5. The above research results are consistent with Rahman's research [65] in the UK which found that companies with a long history of being privately owned tend to be conservative in their perception of socially responsible activities. Senior management has not paid adequate attention to social responsibility disclosure. Confirming hypothesis H1, H2, H3 and H4, we can conclude that company size (β = 0.882), industry sensitivity (β = 0.549), government ownership (β = 0.347) and liquidity (β = 0.093) had a significant effect on the level of corporate social responsibilities disclosure (CSRD).

To test the relationship between CSRD and financial performance, we also conducted regression analysis with model (2) and model (3) to test the hypothesis. Table 6 shows the regression results for ROA and ROE as dependent variables and CSRD as independent variables. The finding points out a positive and significant relationship between CSRD, ROA and ROE. According to the results, there is a highly positive and significant correlation between CSRD and ROA and ROE, with $R^2$ of 72.9% and 64.8%, respectively, and *p*-value = 0.000. This means that a company that performs well in corporate social responsibility activities will introduce a larger profit. This result is in line with the study of McWilliams and Siegel [66] for 524 US firms, Makni et al. [67] for 222 Canadian listed firms and Saleh et al. [68] for 200 Malaysian listed companies. However, the results of this study are contrary to those of Rehman et al. [69] in Pakistan, who found a significant negative relationship between CSRD and financial performance. Additionally, no similarity was found between our results and the study of Buallay et al. [70] for 203 firms listed in six Mediterranean countries, which showed no significant relationship between CSRD and financial performance of the sample firms.

**Table 6.** CSRD effect on financial performance.

| CSRD | ROA | ROE |
|---|---|---|
| Significance/*p*-value | 0.000 | 0.000 |
| Adjusted R-squared | 72.9% | 64.8% |
| Observation | 100 | 100 |

Note: Table 6 presents regression results between CSRD and financial performance at a 1% confidence level.

## 6. Conclusions and Recommendations

This study examines whether the company size, industry sensitivity, government ownership, liquidity and company age key drivers of CSRD among Vietnamese listed enterprises. Second, this study explores the effect of CSRD on companies' financial performance. We used the multiple regression method to test the proposed hypotheses. Company size was found to have the most decisive influence on the level of CSRD, followed by industry sensitivity, government ownership and liquidity. Company age does not seem to affect CSRD. Our empirical findings are in line with the stakeholder theory when Vietnamese listed companies have to disclose social responsibility information to satisfy their interests and requirements. We also found that financial performance is significantly related to CSRD activities. Both ROA and ROE can be explained by the difference in the disclosure of CSR information among Vietnamese listed companies. Social responsibility disclosure information has a positive impact on a company's financial performance.

The study's findings contribute three insightful, practical implications and recommendations. First, our research is helpful to investors by providing an analysis of the relationship between the CSRD and characteristics of listed companies in developing countries. Investors should select large-scale, state-owned and environmentally sensitive industries enterprises that are providing more information about social responsibility, as

the transparency in CSRD is closely related to the financial performance of a business. Second, corporate social responsibility awareness of enterprises in Vietnam should be enhanced. Along with the trend of sustainable development, stakeholders are increasingly focused on corporate social responsibility activities. Therefore, Vietnamese listed companies should use resources to carry out their social responsibility through environmental responsibility, employee responsibility and community responsibility, and become more proactive in disclosing their social responsibility information. This will have a positive effect on the financial performance of enterprises in the long term. Listed companies should be given long-term strategies to apply and disclose social responsibility information with appropriate steps in the integration process towards sustainable development. Third, the government and the State Securities Commission of Vietnam (SSC) should continue to supplement and complete listed companies' legal regulations regarding CSRD. The content of social responsibility information in the annual report provided to stakeholders presently is voluntary, with few criteria (only 15 items) and much important information that has not been announced. The SSC should develop and set up standards of corporate social responsibility information disclosure under international practice, and provide sufficient information to listed companies. These companies can use the GRI standards to prepare a social responsibility report and sustainability report separate from an annual report to increase accountability and enhance the transparency of their contribution to sustainable development.

This study also has several limitations. The study focuses only on CSRD in annual reports, even though these companies may use other media channels such as websites or investor meetings to announce the relevant information. Second, although the data sample was representative of more than 80% of the whole market's transaction value, the number of companies surveyed is small. Using a larger sample is likely to add new insights on CSRD in Vietnam. Third, some factors affecting CSRD have not been studied in the article, such as operating leverage, human rights and anti-corruption issues. Further research can include these factors and employ cross-sectional variations to provide a broader social responsibility report's content and meaning. Moreover, an extensive sample of listed companies could be utilized to evaluate more industry sectors of the economy. Future research that proposes regulatory frameworks for social responsibilities and sustainable development reports could be considered to help stakeholders enhance credibility with the reports and to limit risk awareness among capital providers.

**Author Contributions:** Conceptualization, T.H.N. and Q.T.V.; methodology, T.H.N. and D.M.N.; validation, T.H.N. and H.L.L.; formal analysis, T.H.N. and D.M.N.; resources, T.H.N. and Q.T.V.; writing—original draft preparation, T.H.N., Q.T.V. and H.L.L. All authors have read and agreed to the published version of the manuscript.

**Funding:** This research received no external funding.

**Institutional Review Board Statement:** Not applicable.

**Informed Consent Statement:** Not applicable.

**Data Availability Statement:** Not applicable.

**Conflicts of Interest:** The authors declare no conflict of interest.

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
