# Peer review of "Factors Influencing Corporate Social Responsibility Disclosure and Its Impact on Financial Performance: The Case of Vietnam"

_sustainability, doi:10.3390/su13158197_

Round 1

Reviewer 1 Report

My comments on the  paper - Factors Influencing Corporate Social Responsibility Disclosure and Its Impact on Financial Performance: The Case of Vietnam- are as follows.

The paper presents an interesting analysis and we consider that the research is of interest. However, as it stands, the paper needs revisions.

The abstract is clear, presents the purpose of the paper and the results of the research.

The structure of the paper is coherent.

The research methodology used by the author is adequate for the approached subject.

The references used by the author are appropriate.

The conclusions are significant.

In the introduction it must be stated the added value that the paper brings to the existing academic literature.

We recommend indicating the references for the tables, but also for the figure.

We recommend the development of the discussions in section 5. This is an important part of the paper that is rather underdeveloped. The results are briefly explained. Most often, the author summarizes indicating the relationship between variables. However, we recommend explaining the causes that determined the behaviour of the analysed variables.

Author Response

A Summary of Responses to Reviewers' Reports

Dear Editor and the reviewers,

Thank you very much for the opportunity to revise and resubmit our paper. Reviewers acknowledge the value of the paper and its likely interest to readers. The points made by the Editor and reviewers are very constructive, insightful and worthwhile, and so we have given each suggestion full consideration as detailed below. Using the “Track Changes” function, we have marked the changes within the revised manuscript and highlighted them in yellow.

Thank you very much and all the very best.

Reviewer #1:                                                                                                                               

Point 1: In the introduction it must be stated the added value that the paper brings to the existing academic literature.

Response 1:

Thank you for pointing this out. We agree with this comment and we’ve made the change. Please see page 2, line 21-24 of the revised manuscript.

Point 2: We recommend indicating the references for the tables, but also for the figure.

Response 2: We understand the concern of the reviewer. The authors have added references in the tables and figures to clarify the calculation of indicators in this. Please see page 9 (table 2), page 10 (table 3), page 11 (table 5), page 12 (table 6) of the revised manuscript.

Point 3: We recommend the development of the discussions in section 5. This is an important part of the paper that is rather underdeveloped. The results are briefly explained. Most often, the author summarizes indicating the relationship between variables. However, we recommend explaining the causes that determined the behaviour of the analysed variables

Response 3:

We thank the reviewer for their valuable suggestion regarding the discussions in section 5. We agree with this comment. The authors have added explaining the causes that determined the behaviour of the analysed variables. The corresponding text in the section “Discussion” has also been updated. Please see page 11,12 of the revised manuscript.

Reviewer #2

 1. Introduction

Point 1: This study uses a lot of space to describe the degree of importance CSRD has received. It also mentions that CSRD research is mostly related to annual reports. Therefore, a more comprehensive method is needed to judge the degree of practice of corporate social responsibility, which is affirmative. However, it did not elaborate on the shortcomings or drawbacks of traditional annual reports to observe the implementation of corporate social responsibility.

Response 1:

We agree with this comment. We have clarified the shortcomings of the annual report in providing corporate social responsibility information. Please see page 2, line 16-21 of the revised manuscript.

Point 2: Although the scope of the research is limited, why this article focuses on the relationship between CSRD and financial performance, but does not elaborate.

Response 2: Thank you for this suggestion. It would have been interesting to explore this aspect. However, in our study, Vietnamese enterprises do not publish CSR information according to the standards of the Global Reporting Initiative (GRI). The information about each content of CSR is not enough to test the specific relationship between CSR’s contents (Environment, Employees, Society) and financial performance. So, the study only evaluates the influence of CSRD (combine all three contents) on financial performance.

  1. Theory and Hypotheses

Point 1: This research attempts to analyze and discuss corporate social responsibility based on stakeholder theory and legitimacy theory. However, the title of the article is based on the elements that affect CSRD, and how to select or judge influencing factors. What is the relevant theoretical basis?

Response 1:

Thank you for pointing this out. Our research is based on stakeholder theory and legitimacy theory to select and examine the factors influencing Corporate Social Responsibility Disclosure and Its Impact on the Financial Performance of Vietnamese companies. As each economy experiences different characteristics, a study on Vietnamese firms is necessary to understand the topic in the context of a developing country. There is no issue of the relevant theoretical basis.

Point 2: Another focus of this article is to explore the relationship or impact between CSRD and the company's financial performance. The theoretical basis used does not seem to be clearly stated?

Response 2:

Several prior researchers have undertaken the test for the relationship between CSRD and the company’s financial performance (Please see the [29],[31],[57],[58],[59] …). However, the empirical findings are vague. This is the reason we conduct this research using data from Vietnamese listed companies to clarify the issue.

  1. Methodology

Point 1: This article proposes several factors that will affect CSRD, such as company size, industry sensitivity, government equity ratio, or company’s age, etc. However, what are the reasons for choosing these factors? Even though these factors are positively correlated with CSRD, use The information or data to be corroborated does not seem to be stated

Response 1: All the factors are selected by carefully analysing the previous literature.

Point 2: In addition, Figure 1 on page 6 mainly hopes to propose the possible influence of CSRD on financial performance, but there is no strong argumentation basis or statistical analysis in the relevant discussions. If Vietnam’s local cases are limited, it is recommended that similar markets (such as other ASEAN countries) or markets with similar developments should be selected to find the basis for data and analysis. As an analogy, comparison and analysis should make the discussion in this article more complete.

Response 2:

We thank the reviewer for their valuable suggestion regarding the influence of CSRD on financial performance. We have added some studies about this relationship in the ASEAN region. Please see this change in page 6 (line 3-10 and 18-23) of the revised manuscript.

  1. Results & Discussion

Point 1: The conclusion of this article is that CSRD is closely related to financial performance, and this argument has been discussed and confirmed in many studies. There may be correctness in the direction, but whether there is an absolute positive correlation between the two still needs to be discussed and studied. In particular, the number of samples in this study does not seem to be large enough. Is it possible to confirm that there is an absolute correlation between the two? In addition, the so-called disclosure, what key points or the level of disclosure, seems to also affect financial performance, these should be research The focus is on. Although this article derives the relevant results through the calculation of ROE and ROA, the real influencing factors may need to be further discussed

Response 1:

We acknowledge that there are many studies on relationship between CSRD and financial performance showing negative results, or no relationship. Therefore, we also added this point. Refer to page 12, line 31-35. We agree that the size of samples is not large due to time and budget constraints, which is the limitation of the study and has been presented in the article.

Reviewer #3

Point 1: Line 343–344 for the three formulas (1), (2), (3), are they linear regressions for dependent variables CSRD, ROA, and ROE? If they are, then where are the time series for your independent variables? how to regress without time series data? or you just matching the variable numbers in the formulas. How to make sense of Table 4 and Table 5.

Response 1:

Thank you for pointing this out. Due to financial constraints and limited time to access and collect data of previous years (from 2018 or earlier), we collected data on the annual report of the 100 largest capitalization enterprises (call VN100) in 2019. Therefore, the linear regression equations did not use time-series data. Instead, we used 100 observations at 100 enterprises in the VN100 for all three regression equations. The results of data analysis are presented in Table 4 described the correlation coefficients among independent variables in regression equation 1. The data in Table 5 described the linear regression results between the independent variables (CSRD) and five dependent variables, including SIZE, INDS, GOV, LIQ, AGE.

Reviewer #4

Point 1: Abbreviations are not usually written in abstract. Also it is needed to add scientific problem here. Doubts whether it is sufficient to cite even a few articles at the thought of one sentence. Do  they really write about the same? I mean  [12-15] or  [4,16-22] and etc.

Response 1:

We agreed. We have added the full names of abbreviations in the abstract (highlighted in yellow on page 2 of the revised manuscript). Regarding citations in [12-15] or [4,16-22], we mentioned that these papers used the Stakeholders theory and Legitimacy theory in their research. Other citations related to multiple authors are based on those authors having the same opinion and research results instead of based on similar words or sentences.

Point 2: Although the list of scientific literature is abundant, I would reccomend to add at least a few articles from recent years

Response 2:

Agreed. We have added several recent articles (references number 58,61,71,72)

Point 3: Also lack of analysis of another elements of corporate social responsibility: human rights, anticorruption and etc. Now mostly analysis is based on environmental issues. It should be added and did additional anlysis or written in limitation part.

Response 3:

We agreed with this comment. In this article, we have not studied about human rights, anticorruption, and etc in Vietnamese enterprises. This issue can be considered as one of the limitations of the study. We have added this in the new limitations (see highlighted in page 13).

Please see attached file for revised manuscript. 

Reviewer 2 Report

The study examines the impact of company size, industry sensitivity, Government ownership, liquidity and company age on Corporate Social Responsibility Disclosure (CSRD) in 2019 annual reports of listed companies on the Vietnam stock market. Besides, this study consider the relationship between CSRD and the financial performance measured by ROA and ROE. This study uses 11 descriptive statistics and regression methods to test research hypotheses. The empirical findings show that company characteristics, including firm size, liquidity, Government ownership and environmental industry sensitivity, positively associate with the firms’ CSRD level. From the theoretical structure and research conclusions, this article has contributed. Put forward the following points for the author's reference:

  1. Introduction

(1) This study uses a lot of space to describe the degree of importance CSRD has received. It also mentions that CSRD research is mostly related to annual reports. Therefore, a more comprehensive method is needed to judge the degree of practice of corporate social responsibility, which is affirmative. However, it did not elaborate on the shortcomings or drawbacks of traditional annual reports to observe the implementation of corporate social responsibility.

(2) Although the scope of the research is limited, why this article focuses on the relationship between CSRD and financial performance, but does not elaborate.

  1. Theory and Hypotheses

(1) This research attempts to analyze and discuss corporate social responsibility based on stakeholder theory and legitimacy theory. However, the title of the article is based on the elements that affect CSRD, and how to select or judge influencing factors. What is the relevant theoretical basis?

(2) Another focus of this article is to explore the relationship or impact between CSRD and the company's financial performance. The theoretical basis used does not seem to be clearly stated..

  1. Methodology

(1) This article proposes several factors that will affect CSRD, such as company size, industry sensitivity, government equity ratio, or company’s age, etc. However, what are the reasons for choosing these factors? Even though these factors are positively correlated with CSRD, use The information or data to be corroborated does not seem to be stated.

(2) In addition, Figure 1 on page 6 mainly hopes to propose the possible influence of CSRD on financial performance, but there is no strong argumentation basis or statistical analysis in the relevant discussions. If Vietnam’s local cases are limited, it is recommended that similar markets (such as other ASEAN countries) or markets with similar developments should be selected to find the basis for data and analysis. As an analogy, comparison and analysis should make the discussion in this article more complete.

  1. Results & 5. Discussion

The conclusion of this article is that CSRD is closely related to financial performance, and this argument has been discussed and confirmed in many studies. There may be correctness in the direction, but whether there is an absolute positive correlation between the two still needs to be discussed and studied. In particular, the number of samples in this study does not seem to be large enough. Is it possible to confirm that there is an absolute correlation between the two? In addition, the so-called disclosure, what key points or the level of disclosure, seems to also affect financial performance, these should be research The focus is on. Although this article derives the relevant results through the calculation of ROE and ROA, the real influencing factors may need to be further discussed.

Author Response

(The authors gave the same response as above.)

Reviewer 3 Report

  1. Line 343–344 for the three formulas (1), (2), (3), are they linear regressions for dependent variables CSRD, ROA, and ROE? If they are, then where are the time series for your independent variables? how to regress without time series data? or you just matching the variable numbers in the formulas. How to make sense of Table 4 and Table 5

Author Response

A Summary of Responses to Reviewers' Reports

Dear Editor and the reviewers,

Thank you very much for the opportunity to revise and resubmit our paper. Reviewers acknowledge the value of the paper and its likely interest to readers. The points made by the Editor and reviewers are very constructive, insightful and worthwhile, and so we have given each suggestion full consideration as detailed below. Using the “Track Changes” function, we have marked the changes within the revised manuscript and highlighted them in yellow.

Thank you very much and all the very best.

Reviewer #1:                                                                                                                               

Point 1: In the introduction it must be stated the added value that the paper brings to the existing academic literature.

Response 1:

Thank you for pointing this out. We agree with this comment and we’ve made the change. Please see page 2, line 21-24 of the revised manuscript.

Point 2: We recommend indicating the references for the tables, but also for the figure.

Response 2: We understand the concern of the reviewer. The authors have added references in the tables and figures to clarify the calculation of indicators in this. Please see page 9 (table 2), page 10 (table 3), page 11 (table 5), page 12 (table 6) of the revised manuscript.

Point 3: We recommend the development of the discussions in section 5. This is an important part of the paper that is rather underdeveloped. The results are briefly explained. Most often, the author summarizes indicating the relationship between variables. However, we recommend explaining the causes that determined the behaviour of the analysed variables

Response 3:

We thank the reviewer for their valuable suggestion regarding the discussions in section 5. We agree with this comment. The authors have added explaining the causes that determined the behaviour of the analysed variables. The corresponding text in the section “Discussion” has also been updated. Please see page 11,12 of the revised manuscript.

Reviewer #2 

  1. Introduction

Point 1: This study uses a lot of space to describe the degree of importance CSRD has received. It also mentions that CSRD research is mostly related to annual reports. Therefore, a more comprehensive method is needed to judge the degree of practice of corporate social responsibility, which is affirmative. However, it did not elaborate on the shortcomings or drawbacks of traditional annual reports to observe the implementation of corporate social responsibility.

Response 1:

We agree with this comment. We have clarified the shortcomings of the annual report in providing corporate social responsibility information. Please see page 2, line 16-21 of the revised manuscript.

Point 2: Although the scope of the research is limited, why this article focuses on the relationship between CSRD and financial performance, but does not elaborate.

Response 2: Thank you for this suggestion. It would have been interesting to explore this aspect. However, in our study, Vietnamese enterprises do not publish CSR information according to the standards of the Global Reporting Initiative (GRI). The information about each content of CSR is not enough to test the specific relationship between CSR’s contents (Environment, Employees, Society) and financial performance. So, the study only evaluates the influence of CSRD (combine all three contents) on financial performance.

  1. Theory and Hypotheses

Point 1: This research attempts to analyze and discuss corporate social responsibility based on stakeholder theory and legitimacy theory. However, the title of the article is based on the elements that affect CSRD, and how to select or judge influencing factors. What is the relevant theoretical basis?

Response 1:

Thank you for pointing this out. Our research is based on stakeholder theory and legitimacy theory to select and examine the factors influencing Corporate Social Responsibility Disclosure and Its Impact on the Financial Performance of Vietnamese companies. As each economy experiences different characteristics, a study on Vietnamese firms is necessary to understand the topic in the context of a developing country. There is no issue of the relevant theoretical basis.

Point 2: Another focus of this article is to explore the relationship or impact between CSRD and the company's financial performance. The theoretical basis used does not seem to be clearly stated?

Response 2:

Several prior researchers have undertaken the test for the relationship between CSRD and the company’s financial performance (Please see the [29],[31],[57],[58],[59] …). However, the empirical findings are vague. This is the reason we conduct this research using data from Vietnamese listed companies to clarify the issue.

  1. Methodology

Point 1: This article proposes several factors that will affect CSRD, such as company size, industry sensitivity, government equity ratio, or company’s age, etc. However, what are the reasons for choosing these factors? Even though these factors are positively correlated with CSRD, use The information or data to be corroborated does not seem to be stated

Response 1: All the factors are selected by carefully analysing the previous literature.

Point 2: In addition, Figure 1 on page 6 mainly hopes to propose the possible influence of CSRD on financial performance, but there is no strong argumentation basis or statistical analysis in the relevant discussions. If Vietnam’s local cases are limited, it is recommended that similar markets (such as other ASEAN countries) or markets with similar developments should be selected to find the basis for data and analysis. As an analogy, comparison and analysis should make the discussion in this article more complete.

Response 2:

We thank the reviewer for their valuable suggestion regarding the influence of CSRD on financial performance. We have added some studies about this relationship in the ASEAN region. Please see this change in page 6 (line 3-10 and 18-23) of the revised manuscript.

  1. Results & Discussion

Point 1: The conclusion of this article is that CSRD is closely related to financial performance, and this argument has been discussed and confirmed in many studies. There may be correctness in the direction, but whether there is an absolute positive correlation between the two still needs to be discussed and studied. In particular, the number of samples in this study does not seem to be large enough. Is it possible to confirm that there is an absolute correlation between the two? In addition, the so-called disclosure, what key points or the level of disclosure, seems to also affect financial performance, these should be research The focus is on. Although this article derives the relevant results through the calculation of ROE and ROA, the real influencing factors may need to be further discussed

Response 1:

We acknowledge that there are many studies on relationship between CSRD and financial performance showing negative results, or no relationship. Therefore, we also added this point. Refer to page 12, line 31-35. We agree that the size of samples is not large due to time and budget constraints, which is the limitation of the study and has been presented in the article.

Reviewer #3

Point 1: Line 343–344 for the three formulas (1), (2), (3), are they linear regressions for dependent variables CSRD, ROA, and ROE? If they are, then where are the time series for your independent variables? how to regress without time series data? or you just matching the variable numbers in the formulas. How to make sense of Table 4 and Table 5.

Response 1:

Thank you for pointing this out. Due to financial constraints and limited time to access and collect data of previous years (from 2018 or earlier), we collected data on the annual report of the 100 largest capitalization enterprises (call VN100) in 2019. Therefore, the linear regression equations did not use time-series data. Instead, we used 100 observations at 100 enterprises in the VN100 for all three regression equations. The results of data analysis are presented in Table 4 described the correlation coefficients among independent variables in regression equation 1. The data in Table 5 described the linear regression results between the independent variables (CSRD) and five dependent variables, including SIZE, INDS, GOV, LIQ, AGE.

Reviewer #4

Point 1: Abbreviations are not usually written in abstract. Also it is needed to add scientific problem here. Doubts whether it is sufficient to cite even a few articles at the thought of one sentence. Do  they really write about the same? I mean  [12-15] or  [4,16-22] and etc.

Response 1:

We agreed. We have added the full names of abbreviations in the abstract (highlighted in yellow on page 2 of the revised manuscript). Regarding citations in [12-15] or [4,16-22], we mentioned that these papers used the Stakeholders theory and Legitimacy theory in their research. Other citations related to multiple authors are based on those authors having the same opinion and research results instead of based on similar words or sentences.

Point 2: Although the list of scientific literature is abundant, I would reccomend to add at least a few articles from recent years

Response 2:

Agreed. We have added several recent articles (references number 58,61,71,72)

Point 3: Also lack of analysis of another elements of corporate social responsibility: human rights, anticorruption and etc. Now mostly analysis is based on environmental issues. It should be added and did additional anlysis or written in limitation part.

Response 3:

We agreed with this comment. In this article, we have not studied about human rights, anticorruption, and etc in Vietnamese enterprises. This issue can be considered as one of the limitations of the study. We have added this in the new limitations (see highlighted in page 13).

Please see attached file for revised manuscript. 

Sincerely,

The authors of paper 1207220.

Reviewer 4 Report

Abbreviations are not usually written in abstract. Also it is needed to add scientific problem here. Doubts whether it is sufficient to cite even a few articles at the thought of one sentence. Do  they really write about the same? I mean  [12-15] or  [4,16-22] and etc. Although the list of scientific literature is abundant, I would reccomend to add at least a few articles from recent years. Also lack of analysis of another elements of corporate social responsibility: human rights, anticorruption and etc. Now mostly analysis is based on environmental issues. It should be added and did additional anlysis or written in limitation part. 

Author Response

A Summary of Responses to Reviewers' Reports

Dear Editor and the reviewers,

Thank you very much for the opportunity to revise and resubmit our paper. Reviewers acknowledge the value of the paper and its likely interest to readers. The points made by the Editor and reviewers are very constructive, insightful and worthwhile, and so we have given each suggestion full consideration as detailed below. Using the “Track Changes” function, we have marked the changes within the revised manuscript and highlighted them in yellow.

Thank you very much and all the very best.

Reviewer #1:                                                                                                                            

Point 1: In the introduction it must be stated the added value that the paper brings to the existing academic literature.

Response 1:

Thank you for pointing this out. We agree with this comment and we’ve made the change. Please see page 2, line 21-24 of the revised manuscript.

Point 2: We recommend indicating the references for the tables, but also for the figure.

Response 2: We understand the concern of the reviewer. The authors have added references in the tables and figures to clarify the calculation of indicators in this. Please see page 9 (table 2), page 10 (table 3), page 11 (table 5), page 12 (table 6) of the revised manuscript.

Point 3: We recommend the development of the discussions in section 5. This is an important part of the paper that is rather underdeveloped. The results are briefly explained. Most often, the author summarizes indicating the relationship between variables. However, we recommend explaining the causes that determined the behaviour of the analysed variables

Response 3:

We thank the reviewer for their valuable suggestion regarding the discussions in section 5. We agree with this comment. The authors have added explaining the causes that determined the behaviour of the analysed variables. The corresponding text in the section “Discussion” has also been updated. Please see page 11,12 of the revised manuscript.

Reviewer #2 

  1. Introduction

Point 1: This study uses a lot of space to describe the degree of importance CSRD has received. It also mentions that CSRD research is mostly related to annual reports. Therefore, a more comprehensive method is needed to judge the degree of practice of corporate social responsibility, which is affirmative. However, it did not elaborate on the shortcomings or drawbacks of traditional annual reports to observe the implementation of corporate social responsibility.

Response 1:

We agree with this comment. We have clarified the shortcomings of the annual report in providing corporate social responsibility information. Please see page 2, line 16-21 of the revised manuscript.

Point 2: Although the scope of the research is limited, why this article focuses on the relationship between CSRD and financial performance, but does not elaborate.

Response 2: Thank you for this suggestion. It would have been interesting to explore this aspect. However, in our study, Vietnamese enterprises do not publish CSR information according to the standards of the Global Reporting Initiative (GRI). The information about each content of CSR is not enough to test the specific relationship between CSR’s contents (Environment, Employees, Society) and financial performance. So, the study only evaluates the influence of CSRD (combine all three contents) on financial performance.

  1. Theory and Hypotheses

Point 1: This research attempts to analyze and discuss corporate social responsibility based on stakeholder theory and legitimacy theory. However, the title of the article is based on the elements that affect CSRD, and how to select or judge influencing factors. What is the relevant theoretical basis?

Response 1:

Thank you for pointing this out. Our research is based on stakeholder theory and legitimacy theory to select and examine the factors influencing Corporate Social Responsibility Disclosure and Its Impact on the Financial Performance of Vietnamese companies. As each economy experiences different characteristics, a study on Vietnamese firms is necessary to understand the topic in the context of a developing country. There is no issue of the relevant theoretical basis.

Point 2: Another focus of this article is to explore the relationship or impact between CSRD and the company's financial performance. The theoretical basis used does not seem to be clearly stated?

Response 2:

Several prior researchers have undertaken the test for the relationship between CSRD and the company’s financial performance (Please see the [29],[31],[57],[58],[59] …). However, the empirical findings are vague. This is the reason we conduct this research using data from Vietnamese listed companies to clarify the issue.

  1. Methodology

Point 1: This article proposes several factors that will affect CSRD, such as company size, industry sensitivity, government equity ratio, or company’s age, etc. However, what are the reasons for choosing these factors? Even though these factors are positively correlated with CSRD, use The information or data to be corroborated does not seem to be stated

Response 1: All the factors are selected by carefully analysing the previous literature.

Point 2: In addition, Figure 1 on page 6 mainly hopes to propose the possible influence of CSRD on financial performance, but there is no strong argumentation basis or statistical analysis in the relevant discussions. If Vietnam’s local cases are limited, it is recommended that similar markets (such as other ASEAN countries) or markets with similar developments should be selected to find the basis for data and analysis. As an analogy, comparison and analysis should make the discussion in this article more complete.

Response 2:

We thank the reviewer for their valuable suggestion regarding the influence of CSRD on financial performance. We have added some studies about this relationship in the ASEAN region. Please see this change in page 6 (line 3-10 and 18-23) of the revised manuscript.

  1. Results & Discussion

Point 1: The conclusion of this article is that CSRD is closely related to financial performance, and this argument has been discussed and confirmed in many studies. There may be correctness in the direction, but whether there is an absolute positive correlation between the two still needs to be discussed and studied. In particular, the number of samples in this study does not seem to be large enough. Is it possible to confirm that there is an absolute correlation between the two? In addition, the so-called disclosure, what key points or the level of disclosure, seems to also affect financial performance, these should be research The focus is on. Although this article derives the relevant results through the calculation of ROE and ROA, the real influencing factors may need to be further discussed

Response 1:

We acknowledge that there are many studies on relationship between CSRD and financial performance showing negative results, or no relationship. Therefore, we also added this point. Refer to page 12, line 31-35. We agree that the size of samples is not large due to time and budget constraints, which is the limitation of the study and has been presented in the article.

Reviewer #3

Point 1: Line 343–344 for the three formulas (1), (2), (3), are they linear regressions for dependent variables CSRD, ROA, and ROE? If they are, then where are the time series for your independent variables? how to regress without time series data? or you just matching the variable numbers in the formulas. How to make sense of Table 4 and Table 5.

Response 1:

Thank you for pointing this out. Due to financial constraints and limited time to access and collect data of previous years (from 2018 or earlier), we collected data on the annual report of the 100 largest capitalization enterprises (call VN100) in 2019. Therefore, the linear regression equations did not use time-series data. Instead, we used 100 observations at 100 enterprises in the VN100 for all three regression equations. The results of data analysis are presented in Table 4 described the correlation coefficients among independent variables in regression equation 1. The data in Table 5 described the linear regression results between the independent variables (CSRD) and five dependent variables, including SIZE, INDS, GOV, LIQ, AGE.

 Reviewer #4

Point 1: Abbreviations are not usually written in abstract. Also it is needed to add scientific problem here. Doubts whether it is sufficient to cite even a few articles at the thought of one sentence. Do  they really write about the same? I mean  [12-15] or  [4,16-22] and etc.

Response 1:

We agreed. We have added the full names of abbreviations in the abstract (highlighted in yellow on page 2 of the revised manuscript). Regarding citations in [12-15] or [4,16-22], we mentioned that these papers used the Stakeholders theory and Legitimacy theory in their research. Other citations related to multiple authors are based on those authors having the same opinion and research results instead of based on similar words or sentences.

Point 2: Although the list of scientific literature is abundant, I would reccomend to add at least a few articles from recent years

Response 2:

Agreed. We have added several recent articles (references number 58,61,71,72)

Point 3: Also lack of analysis of another elements of corporate social responsibility: human rights, anticorruption and etc. Now mostly analysis is based on environmental issues. It should be added and did additional anlysis or written in limitation part.

Response 3:

We agreed with this comment. In this article, we have not studied about human rights, anticorruption, and etc in Vietnamese enterprises. This issue can be considered as one of the limitations of the study. We have added this in the new limitations (see highlighted in page 13).

Please see attached file for revised manuscript

Sincerely,

The authors of paper 1207220

Round 2

Reviewer 2 Report

1. In response to the questions and suggestions raised, we have made perfect replies;
2. This topic is worth continuing to discuss, and the author is encouraged to have more in-depth research in the future.

This manuscript is a resubmission of an earlier submission. The following is a list of the peer review reports and author responses from that submission.